# An Automatic DWI/FLAIR Mismatch Assessment of Stroke Patients

**DOI:** 10.3390/diagnostics14010069

**Published:** 2023-12-27

**Authors:** Jacob Johansen, Cecilie Mørck Offersen, Jonathan Frederik Carlsen, Silvia Ingala, Adam Espe Hansen, Michael Bachmann Nielsen, Sune Darkner, Akshay Pai

**Affiliations:** 1Department of Computer Science, University of Copenhagen, 2100 Copenhagen, Denmark; darkner@di.ku.dk (S.D.); ap@cerebriu.com (A.P.); 2Cerebriu A/S, 1434 Copenhagen, Denmark; si@cerebriu.com; 3Department of Clinical Medicine, University of Copenhagen, 2100 Copenhagen, Denmark; jonathan.frederik.carlsen@regionh.dk (J.F.C.); adam.espe.hansen@regionh.dk (A.E.H.); michael.bachmann.nielsen@regionh.dk (M.B.N.); 4Department of Radiology, Copenhagen University Hospital, 2100 Copenhagen, Denmark

**Keywords:** DWI/FLAIR mismatch, ischemic stroke, wake-up stroke, r-tPA, MRI

## Abstract

DWI/FLAIR mismatch assessment for ischemic stroke patients shows promising results in determining if patients are eligible for recombinant tissue-type plasminogen activator (r-tPA) treatment. However, the mismatch criteria suffer from two major issues: binary classification of a non-binary problem and the subjectiveness of the assessor. In this article, we present a simple automatic method for segmenting stroke-related parenchymal hyperintensities on FLAIR, allowing for an automatic and continuous DWI/FLAIR mismatch assessment. We further show that our method’s segmentations have comparable inter-rater agreement (DICE 0.820, SD 0.12) compared to that of two neuro-radiologists (DICE 0.856, SD 0.07), that our method appears robust to hyper-parameter choices (suggesting good generalizability), and lastly, that our methods continuous DWI/FLAIR mismatch assessment correlates to mismatch assessments made for a cohort of wake-up stroke patients at hospital submission. The proposed method shows promising results in automating the segmentation of parenchymal hyperintensity within ischemic stroke lesions and could help reduce inter-observer variability of DWI/FLAIR mismatch assessment performed in clinical environments as well as offer a continuous assessment instead of the current binary one.

## 1. Introduction

Since the approval of recombinant tissue-type plasminogen activator (r-tPA) as a treatment for ischemic stroke by the FDA in 1996 and EMA in 2002, patients eligible for treatment have been selected based on time since symptom onset. The current guidelines in the EU [1] suggest eligibility for treatment up to 4.5 h after onset and are based on the European Cooperative Acute Stroke Study (ECASS) III study [2]. Recent studies, such as Wei et al. and Schwa et al. [3,4], have shown that a subset of patients outside the suggested time window may still benefit from r-tPA treatment, which has raised the question of whether or not the current guidelines are excluding too many patients. Furthermore, the time since symptom onset guideline is not applicable in patients with unknown onset time, which accounts for roughly one in three ischemic stroke patients [5]. This has led to an increased interest in whether other criteria can serve as an alternative or supplemental factor for determining treatment eligibility. One MRI-based criterion, which has received substantial interest in recent years, is the DWI/FLAIR mismatch, commonly used to determine eligibility for unknown onset time patients, following the WAKE-UP trial by Thomalla et al. [6]. The criteria are based on determining whether or not a patient with a visible ischemic lesion on DWI has a parenchymal hyperintensity on FLAIR imaging in the same location. Patients with no parenchymal hyperintensity on FLAIR, referred to as a DWI/FLAIR mismatch, are deemed eligible for treatment under this criterion. While other approaches to identify unknown onset time patients who might benefit from r-tPA treatment exist, such as non-contrast CT, CT-perfusion, and penumbral MRI [7], we chose to focus on DWI/FLAIR mismatch as it is the only criterion used in the hospital systems where the research originated and this is the criteria where expert knowledge and data are avaliable.

One major challenge associated with DWI/FLAIR mismatch in ischemic stroke patients is splitting the patients into one of two subgroups, mismatch or no-mismatch, since the evolution of parenchymal hyperintensities on FLAIR happens continuously over a period of several hours. Therefore, determining whether patients have a mismatch becomes subjective to the neuro-radiologist in question, leading to high inter-observer variability [8]. Recent studies related to DWI/FLAIR mismatch assessment have all focused on classifying whether or not a patient is within the 4.5 h treatment window [9,10,11,12,13], which may not be a favorable approach when it comes to clinical usability [14]. While deep learning models have shown promising results in both general [15,16,17] and medical segmentation tasks [18,19,20], they rely heavily on high quality labeled data for training. To our best knowledge, no open-source dataset exists with parenchymal hyperintensities labeled on FLAIR imaging. This means that applying deep learning models for segmenting parenchymal hyperintensities on FLAIR imaging would require creating such a dataset to begin with. While this is possible, albeit tedious, such a dataset would still suffer from the problem of high inter-rater variability amongst the medical experts creating the ground truth segmentation labels. Out of all the studies we found, only one by Zhu et al. [13] directly segments the parenchymal hyperintensity on FLAIR, achieving a moderate DICE score of 0.647 when compared to manual ground truth segmentations.

We hypothesize that it is possible to automate the DWI/FLAIR mismatch assessment using a simple approach, which segments the parenchymal hyperintensity on FLAIR based on the way actual assessments are currently made. No such model, to the best of our knowledge, currently exists. We expect a simple method based on radiological principles to perform well as well as be robust and generalize well to unseen data, as it should be much less prone to overfitting. This leads us to the following aims: (1) develop an automatic method for segmenting parenchymal hyperintensities within ischemic stroke lesions on FLAIR, allowing us to calculate a continuous DWI/FLAIR mismatch ratio; (2) compare the inter-rater segmentation agreement between the proposed method and neuro-radiologists; and (3) correlate the output of the method with the DWI/FLAIR mismatch assessment made for a group of unknown symptom onset patients.

## 2. Materials and Methods

### 2.1. Study Sample

This retrospective study uses three anonymized datasets. The ISLES 2022 challenge dataset [21] contains 250 publicly available scans from stroke patients obtained at three different stroke centers in Germany. The ISLES team has obtained ethical approval in accordance with the 1964 Declaration of Helsinki from all participating centers and we did not obtain further ethical approval. The dataset includes scans from both 3 T Phillips and Siemens scanners as well as 1.5 T Siemens scanners. A full list of parameters for the dataset, including scanner and acquisition parameters, can be found here: https://www.nature.com/articles/s41597-022-01875-5, (accessed on 1 December 2023). Segmentations were performed on 33 out of 45 selected samples, drawn from the ISLES dataset consecutively. Samples were excluded if judged to be post-intervention or hemorrhagic, as neither of such patients are eligible for r-tPA treatment. Exclusions were performed by a senior neuro-radiologist with 10 years of experience (Reader1). These samples serve to compare the agreement of the algorithms segmentation, with the inter-agreement rate of neuro-radiologists. Segmentations of parenchymal hyperintensity on FLAIR were performed independently by two neuro-radiologists, one senior (Reader1) and one in training (Reader2), using 3D Slicer (https://www.slicer.org/, (accessed on 1 November 2023)). In samples where the inter-agreement DICE score was below 0.7, the segmentations made by the neuro-radiologist-in-training (Reader2) were manually revised by a third neuro-radiologist with 25 years of experience. We have released these labels at OSF (https://osf.io/fc786/?view_only=46193dd7e209424cb41e9831df95c9d5, (accessed on 23 December 23)) for open use. Secondly, we obtained a Wake-up dataset from Univeristy Hospital of Copenhagen in order to correlate our automatic DWI/FLAIR assessment to those made in a real-world clinical setting. We obtained ethical approval from the Danish National Center for Ethics, which waived the right to informed consent. This dataset consists of two consecutive cohorts obtained from two different scanner protocols used at the hospital, one with DWI being a single-shot sequence and the other with DWI being a two-shot sequence. All other scanner parameters were the same. From this cohort we excluded patients with no visible DWI stroke lesion (Figure 1), ending up with 51 unknown onset patients (34 male, age range 52–90, average 71.1; 19 female, age range 36–87, average 71.2; overall average age 71.1), of which 16 were assessed to have a DWI/FLAIR mismatch. All scans were obtained from a 1.5 T GE Healthcare MRI scanner.

Lastly, we used a training dataset from Oslo University Hospital containing three acute (scanned less than 24 h since onset) stroke patients (1 male, age 71, and 2 females, ages 64, 75) with segmentations of the ischemic stroke on DWI and the corresponding parenchymal hyperintensity segmentation on FLAIR available. This dataset was used to calibrate the hyper-parameters of our algorithm. This dataset was ethically approved by the Norwegian Regional Committees for Medical and Health Research Ethics with informed consent waived. These scans were obtained on a 1.5 T Siemens scanner. All images used were retrieved, de-identified, and then converted to Neuroimaging Informatics Technology Initiative format.

### 2.2. Automated FLAIR Segmentations

Our proposed method segments hyper-intensities on FLAIR based on three key features: (1) a general region of interest, defined as the visible ischemic lesion obtained on DWI and projected onto the FLAIR image; (2) the ratio between FLAIR intensity within the region of interest and a reference region on the contralateral side of the brain; and (3) the mean and standard deviation of the intensities in the brain obtained on FLAIR excluding the region of interest and ventricles.

The region of interest was determined by segmenting the ischemic stroke lesion on the DWI and then projecting the segmentation onto the FLAIR image. The DWI and FLAIR images were registered to each other using their affine matrices prior to projection. We note that our method requires a DWI segmentation of the stroke lesion. Studies have shown that automatic segmention of ischemic stroke on DWI is possible at a level comparable to that of neuro-radiologists [22,23]. When we evaluated the segmentation accuracy of the algorithm on the ISLES dataset, we used the manual DWI segmentations available for the dataset as the region of interest. This allowed us to evaluate the algorithm for FLAIR segmentations in an ideal setting with regards to region of interest. When we assessed the correlation between the output of the algorithm and the clinical assessment on the Wake-up dataset, a commercially available U-Net [19], trained to segment stroke lesions on DWI was used to segment the ischemic stroke lesion. However, any model that can reliably segment stroke lesions on DWI could be used along with our model, including newer transformer models, which are currently considered state of the art in regards to many segmentation tasks [24]. While our method is impacted by the quality of the DWI stroke segmentation, it is outside the scope of this paper to further explore different DWI stroke segmentation models.

The contralateral FLAIR intensity information was obtained by mirroring the brain across the plane that separates the left and right brain hemisphere. The mirrored brain is then registered to the non-mirrored one. In practice, this was performed using translation and rotation in order to minimize the absolute difference between the original and mirrored brain. Lastly, the mean and standard deviation of the intensities was obtained for each 2D slice (2D plane perpendicular to the acquisition direction of the scan) in the FLAIR image. These measures were obtained on an area which included the entire brain, but without the ventricles and cerebrospinal fluid. In practice, this area was found using the automatic tool SynthSeg [25], which segments the brain into different regions, including ventricals and cerebrospinal fluid. In our approach, voxels were segmented if they fulfilled at least one of the following two criteria: (1) The relative intensity of the voxel compared to the contralateral side was above 1.15 and the intensity of the voxel was at least as bright as the mean intensity plus 0.25 standard deviations of the intensity of the 2D slice. Or (2), the intensity of the voxel was brighter than the mean plus 1.25 standard deviations of the intensity of the 2D slice.

The criteriion for the contrateral ratio was based on previous studies which explore the contralateral signal intensity ratio optimal for identifying DWI/FLAIR mismatch. These studies suggest a ratio of between 1.07 and 1.15 [26,27,28]. The optimal ratio, along with the multipliers for the standard deviation, was chosen to maximize the overlap on our training dataset. However, since the size of the training dataset was only 3 patients, we assessed the impact on the FLAIR segmentation when these parameters were varied in order to test the robustness and generalizability of our method. Furthermore, since all the papers focused solely on the contralateral ratio as a DWI/FLAIR mismatch identifier, we also tested a version of the algorithm without the multipliers. We note our FLAIR segmentation method runs in less than 5 s (tested on an Intel i5-9600k 3.7 GHz processor (Santa Clara, CA, USA) and the U-Net and Synthseg segmentation run in less than 30 s (tested on a Nvidia RTX 2070 (Santa Clara, CA, USA)), making the worst case scenario runtime 1 min 5 s after image acquisition. However, since U-Net and Synthseg can be run on the DWI only, if the DWI is acquired prior to the FLAIR image, the method can run in almost real time.

For criterion (1), the added condition that the intensity of the voxel should be brighter than a certain threshold was added to prevent segmentation of iso-intense voxels, where the contralateral voxel is hypo-intense. Criterion (2) was used to negate some situations where the contralateral information was not useful and thus criterion (1) would fail. Such situations could arise if strokes are located across the mirroring plane, where part of the contralateral information comes from the stroke itself, as well as strokes where the contralateral site is hyperintense, for example, areas around the ventricles prone to age-related white matter hyperintensities. While criterion (2) can negate some problematic situations, it is possible for a stroke to have poor contralateral information but not fulfill criterion (2), leading to poor segmentation. Such situations can be split into two different scenarios, one where the stroke is located across the mirroring plane causing poor contralateral information and one where the contralateral part is hyperintense but not part of the stroke itself. It is possible to automatically detect the first scenario, where the contralateral information comes from the stroke itself. Since the location of the mirroring plane is known, we can check if the area of interest (DWI segmentation) overlaps itself when mirrored. In the following analysis, we used a threshold of 20% overlap to exclude cases. Meaning that if the volume of overlap is larger than 20% of the total area of interest, we considered the case unreliable and excluded it. In order to negate the second situation, where the stroke does not overlap itself but the contralateral side is hyperintense, we obtained an intensity measure along with our DWI/FLAIR mismatch ratio. The intensity measure was a ratio of intensity between the entire area of interest and the rest of the brain on the FLAIR image, again excluding ventricals and cerebrospinal fluid. Here, we expected such strokes to have a small DWI/FLAIR mismatch ratio since the FLAIR segmentation was unreliable but a large intensity measure and since the area of interest should still have high intensity compared to the rest of the brain.

A flowchart of the proposed algorithm can be found in Figure A1 in Appendix A.

### 2.3. Statistical Analysis

For the ISLES dataset with segmentations available, we analyzed the DICE score between the algorithm and the two neuro-radiologists’ segmentations as well as the inter-rater DICE score using a one-way ANOVA as well as a Tukey Honest Significant Difference test. The goal was to see how the algorithm performed compared to the inter-rater agreement. Furthermore, the robustness of the algorithm was examined by checking the impact on the DICE score when varying the ratio threshold and standard deviation multipliers on this dataset, which was performed using the same two tests, with the neuro-radiologists’ segmentations as ground truth labels. Furthermore, we tested a version of the method that only used the contralateral ratio as a criteria for segmentation, which was the approach in other studies [26,27,28]. This was performed using a *t*-test, with the two neuro-radiologists’ segmentations as ground truth labels. For these tests, we did not test the correctness of the intensity measure, as we had no ground truth to compare it to.

For the Wake-up dataset, we wanted to determine the correlation between the output of the algorithm and DWI/FLAIR mismatch evaluation performed for the unknown symptom onset patients. Here, we calculated the Point-Biserial Correlation Coefficient (PBCC) between our measures (DWI/FLAIR mismatch ratio and intensity measure) and the mismatch assessment performed at the hospital. The Point-Biserial Correlation Coefficient allows for a calculation of correlation between a continuous and a binary variable. Furthermore, we performed a *t*-test to check if the mean DWI/FLAIR mismatch ratio as well as the intensity measure were different for the two groups (mismatch, no mismatch). All conclusions were based on a 5% significance level.

## 3. Results

### 3.1. Inter-Rater Agreement

Image-weighted DICE scores for the FLAIR segmentations, were computed for each possible pair combination on the ISLES 2022 dataset (Table 1).

The agreement was highest between the two neuro-radiologists and lowest between the neuro-radiologist-in-training (Reader2) and the algorithm. A one-way ANOVA showed that the means between the three pairs were different (p<0.02) and a Tukey Honest Significant Difference test showed that the mean between Reader1/Reader2 and Reader1/Algo could not be assumed to be different (p>0.4), while the rest of the pairs could (p<0.04). Statistics for the segmentations can be seen in Table 2. A comparison of segmentations between the two neuro-radiologists showed that Reader1 had a tendency to segment more volume, while Reader2 had a tendency to segment less volume in all cases, and the method segmented more volume than both neuro-radiologists in 27 out of 33 cases.

An analysis of the method’s robustness to the ratio and multiplier choice was conducted. Here, the two multiplier parameters were adjusted to ±20% of the original value and the ratio parameter was adjusted by ±0.08, with the lower bound corresponding to the 1.07 ratio mentioned in Section 2.2. A one-way ANOVA test was conducted on both of the $3^{3}$ combinations, stemming from using either Reader1s or Reader2s segmentations as the ground truth for DICE score computation. Independent of which neuro-radiologist’s segmentations were used, the test did not show a difference in means regardless of hyper-parameter choice (p>0.999). Furthermore, we tested a method which used only the contralateral intensity ratio as a criterion for segmentation. We used both 1.07 and 1.15 as ratios and compared it to our suggested method, using both of the neuro-radiologists segmentations as ground truth. Here the best DICE scores were achieved using our method in all cases. A *t*-test rejected the hypothesis that the mean DICE score was the same between the our method and the algorithm with only the contralateral information in all cases (p>0.08).

### 3.2. DWI/FLAIR Mismatch Agreement

We ran the automated DWI segmentation as well as our FLAIR segmentation method on the Wake-up dataset. Out of the 51 cases, 4 cases were excluded due to the automatic DWI segmentation missing the stroke completely (Figure 2). A quantitative comparison between these patients with missed DWI segmentation and the rest of the cohort showed that the missed lesions all had subtle DWI lesions in regard to intensity. Furthermore, 1 case was excluded due to the stroke being located across the mirroring plane, with more than 20% of the contralateral information being from the stroke itself (Figure 3).

We correlated the clinical mismatch assessment obtained from electronic patient records with our two measures, the DWI/FLAIR mismatch ratio ranging from 0 (complete mismatch) to 1 (complete match) and the intensity ratio measure. A plot of these measures along with the mismatch assessment can be seen in Figure 4.

The 6 numerated cases can be seen in Figure 5. Case 1 is a situation where both the area of interest and the contralateral side are slightly hyperintensive, leading to a small DWI/FLAIR mismatch ratio but large intensity measure and a no-mismatch assessment in the clinic. In Case 2, part of the contralateral side is hyperintense, which may lead to slight under-segmentation. However, the contralateral hyperintensity is not large enough to be clear from the intensity measure. Cases 3 and 4 correspond to two cases where the patients are deemed to be mismatch cases at the hospital, but our measures disagree. Case 5 shows a mismatch assessed case where the measures agree. Case 6 shows a no-mismatch assessed case where the measures and the assessment agree. Furthermore, the case shows the relevance of our criterion that the intensity be brighter than the mean plus 0.25 standard deviations, as part of the contralateral side is hypo-intense.

The Point-Biserial Correlation Coefficient was calculated for the two measures and the binary mismatch assessment. The calculations showed positive and relevant correlation between both the DWI/FLAIR mismatch ratio (PBCC=0.60, p<0.00002) and the intensity measure (PBCC=0.42, p<0.004). This indicates that a higher value of both measures correlates with a no-mismatch assessment. Furthermore, we tested if the mean of the DWI/FLAIR mismatch ratio and the average contralateral intensity ratio was different between the two groups, using a *t*-test. The *t*-test showed a difference in DWI/FLAIR mismatch ratio between the two groups (p<0.0002) as well as the intensity measure (p<0.004). Volume statistics for the Wake-up dataset can be seen in Table 3.

## 4. Discussion

In our study, we have proposed an automatic method for segmenting parenchymal hyperintensity on FLAIR in areas affected by ischemic stroke. We achieved reasonable agreement between the method and neuro-radiologists (DICE >0.715) and better agreement when compared to the only other study found which evaluated this measure (DICE =0.647) [13]. Furthermore, we compared the outcome of the method with the DWI/FLAIR mismatch assessment made at admission time for 46 patients. Here we showed positive and statistical significant correlation for both the DWI/FLAIR mismatch ratio and the intensity measure using the Point-Biserial Correlation Coefficient (PBCC>0.42, p<0.004) and different mean structure in both measures between the two groups using a *t*-test (p<0.004). The study showed that the segmentation method was robust to different hyper-parameter choices, suggesting that the DWI/FLAIR mismatch ratio will generalize well across different scanners and acquisition protocols, as small changes in intensity did not effect the outcome significantly, something which is a large concern when deploying automatic methods in hospital settings [29,30]. Furthermore, the small size of the training dataset also implies that the method should generalize well to unseen data as it is highly unlikely that the training dataset is varied enough to cover all the different patients seen in the two test datasets. The indication of generalizability, paired with the statistical significance of the measures, when comparing the mismatch and no-mismatch patients, suggests that the method could potentially be used to acquire non-subjective DWI/FLAIR mismatch assessments. This would allow for the removal of or reduction in subjectiveness in the current DWI/FLAIR mismatch assessment, which is large problem in both clinical settings [31] as well as when performing and evaluating clinical trials [32]. Furthermore, the method suggested allows for an explanation of why segmentations are produced the way they are, as opposed to most AI models, which do not allow for such explanability. This problem is often referred to as white-box vs. black-box modeling and is a large concern in cases where models are used to make critical choices, not only in healthcare [33]. Lastly, having access to an objective and robust DWI/FLAIR mismatch assessment might allow hospitals to better assess patients eligibility for r-tPA treatment.

While the results seem promising, it should be noted that the Wake-up dataset still showed cases where the measures and assessment seemed to disagree. This, combined with the relatively small size of the dataset, makes it hard to draw robust conclusions about the measure’s eligibility in a clinical setting. Furthermore, since the Wake-up dataset stems from only one hospital, a larger study involving more patients and hospitals would be a logical next step to test the suggested methods clinical eligibility. We also saw that the method had a tendency to over-segment parenchymal hyperintensities on FLAIR. This can most likely be corrected by including more hyper-acute patients in the training set, leading to a better lower threshold of which voxels to segment.

While not the focus of the study, it should also be noted that the automatic DWI segmentation algorithm missed 4 out of 51 patients, all with subtle changes on DWI. Since our approach relies on a reasonable DWI segmentation of the stroke, it is important to validate the DWI segmentation performance before relying on results from our method.

In conclusion, we propose a method for automatically segmenting parenchymal hyperintensity within ischemic stroke lesions and showed that the method’s segmentations are comparable with those made by neuro-radiologists. We showed that the method was robust to hyper-parameter choices, suggesting it should be able to generalize well regardless of scanner or acquisition protocol. Lastly, we showed that the method’s obtained measures correlated with mismatch assessment performed in a real world clinical setting. This suggests that the method could be used to obtain a continuous DWI/FLAIR mismatch ratio in an objective manner, circumventing the major issues related to DWI/FLAIR mismatch assessment, subjectiveness of the reader, and forcing a binary assessment of a non-binary problem.

## Figures and Tables

**Figure 1 diagnostics-14-00069-f001:**
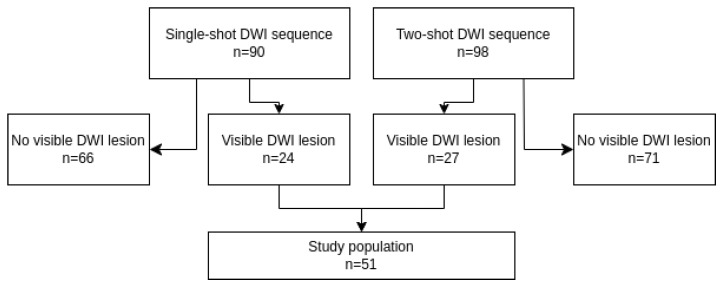
Inclusion/exclusion flowchart for our Wake-up dataset, consisting of two consecutive cohorts stemming from different DWI acquisition protocols.

**Figure 2 diagnostics-14-00069-f002:**
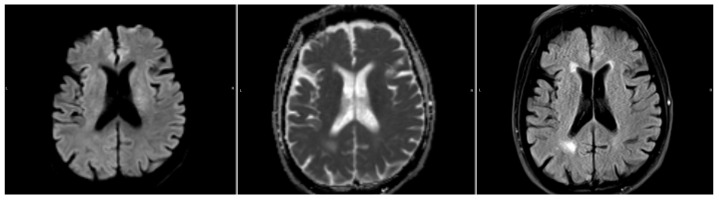
A sample where the automatic DWI segmentation has missed the stroke, located next to the ventricle. From left to right: DWI (b1000), ADC, and FLAIR.

**Figure 3 diagnostics-14-00069-f003:**
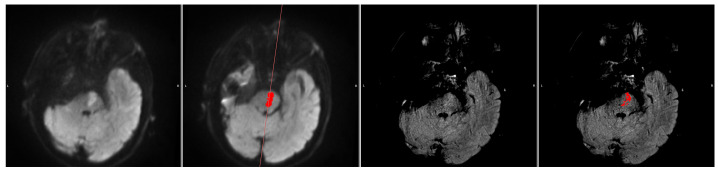
A sample with stroke across mirroring plane. From left to right: DWI (b1000), DWI (b1000) with segmentation (red area) and mirror plane (red line), FLAIR, and FLAIR with segmentation (red area). The ischemic stroke is located in such a way that the contralateral information comes from the stroke itself, making the automatic FLAIR segmentation unreliable.

**Figure 4 diagnostics-14-00069-f004:**
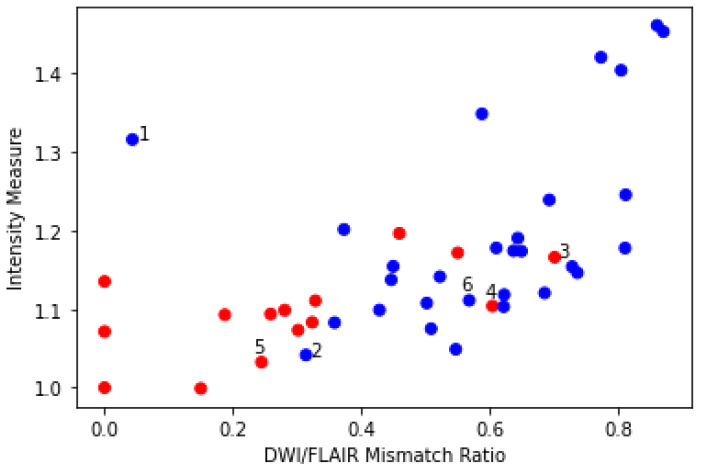
Scatter plot showing the relation between the DWI/FLAIR mismatch ratio, intensity measure, and DWI/FLAIR mismatch assessment for each patient from the Wake-up dataset. Red dots correspond to patients assessed to have a DWI/FLAIR mismatch at hospital admission, while blue dots indicate patients assessed to have no-mismatch at admission. Cases 1–6 can be seen in Figure 5.

**Figure 5 diagnostics-14-00069-f005:**
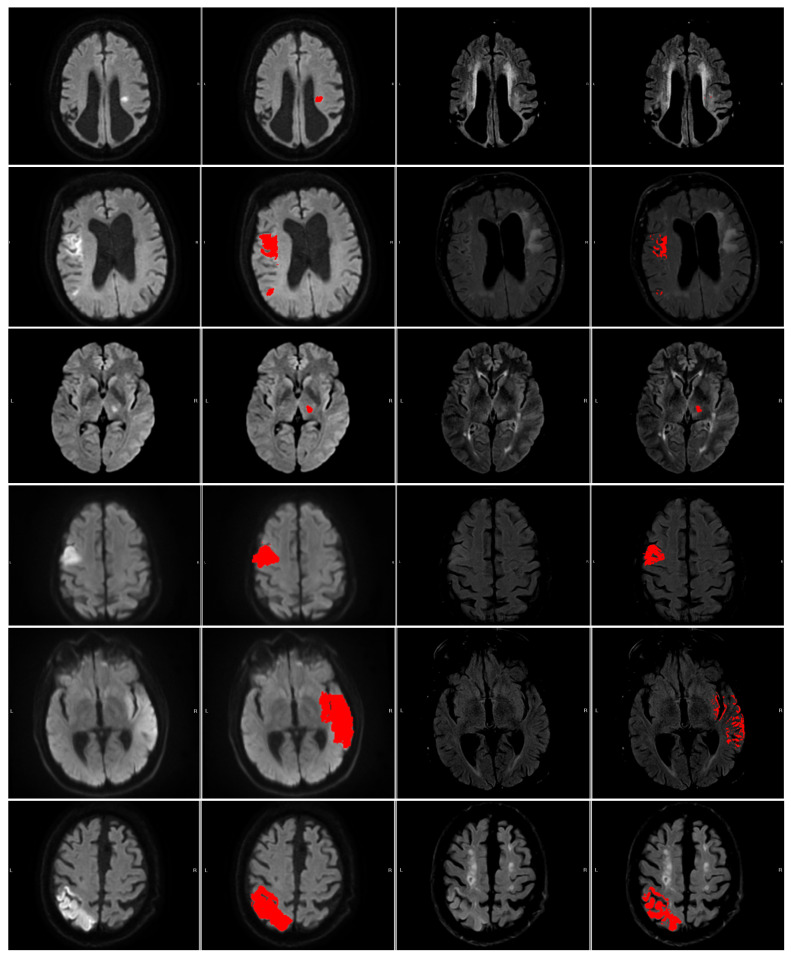
The numbered outlier cases from Figure 4. The cases are presented in ascending order 1 (**top**) through 6 (**bottom**). All cases are shown as (left to right) DWI (b1000), DWI (b1000) with segmentation (red area), FLAIR, and FLAIR with segmentation (red area).

**Table 1 diagnostics-14-00069-t001:** Image-weighted DICE scores for each possible pair of the FLAIR segmentations.

	Reader1/Reader2	Reader1/Algo	Reader2/Algo
Dice (SD)—Image weighted	0.856 (0.07)	0.820 (0.12)	0.749 (0.15)
Worst Dice	0.7	0.444	0.4
Best Dice	1	0.961	0.932

**Table 2 diagnostics-14-00069-t002:** Statistics for the ISLES 2022 FLAIR segmentations. Minimum and maximum number of voxels segmented, minimum and maximum ratio of voxels segmented (FLAIR volume/DWI volume), and average and standard deviation of number of voxels segmented.

	Min Volume	Max Volume	Min Ratio	Max Ratio	Avg. Volume (SD)
DWI	18	11634	N/A	N/A	1388 (2245)
Reader2	6	8700	12.5%	81.2%	843 (1606)
Reader1	9	7661	14%	90.2%	905 (1509)
Algo	14	9596	31.1%	98.2%	1111 (1834)

**Table 3 diagnostics-14-00069-t003:** Statistics for the wake-up segmentations. Minimum and maximum number of voxels segmented, minimum and maximum ratio of voxels segmented (FLAIR volume/DWI volume), and average and standard deviation of number of voxels segmented.

	Min Volume	Max Volume	Min Ratio	Max Ratio	Avg. Volume (σ)
DWI	8	31,095	N/A	N/A	3587 (6062)
FLAIR	0	13,980	0%	87.1%	1735 (2804)

## Data Availability

The ISLES Challenge 2022 dataset is publically avaliable at https://www.isles-challenge.org/ (accessed on 1 December 2023). We have released FLAIR segmentation labels for part of this dataset for open use; they can be found at https://osf.io/fc786/?view_only=46193dd7e209424cb41e9831df95c9d5 (accessed on 23 December 2023). All other data used in the study are not approved for publication due to patient privacy.

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
