# Peer review of "An Automatic DWI/FLAIR Mismatch Assessment of Stroke Patients"

_diagnostics, 2023, doi:10.3390/diagnostics14010069_

Round 1

Reviewer 1 Report

Comments and Suggestions for Authors

1. I think that the methods were trained with image data from only three patients, which is too small a number. (Lines 95~97) The authors described a method to overcome these limitations. It is unknown what the race and age of the patient group in the research data used were, and it is unknown how long it took after symptoms occurred to obtain the imaging test results. Considering this, it seems appropriate to train with a larger number of data from a wider range of ages.

2. In Lines 132-135, the first is thought to be a standard that refers to previous research. For the second, is there a standard or reference material for this?

3. In Figure 4, blue dots are described as meaning no-mismatch. However, on the x-axis below, the DWI/FLAIR mismatch ratio ranges from 0 to 0.8. If there is no mismatch, shouldn't the ratio value be 1?

Author Response

Dear reviewer,

Thank you for taking the time to provide valuable feedback to our article! 

Our comments are;

  1. While the training set is indeed small, we consider this one of the strengths of our method. The fact that we can generalize in a statistical significant way to the two other datasets, as well as show robustness to hyperparameter choices, given our small training dataset, suggests that the method is able to generalize well / is robust / has not overfit the training data. - We could further comment on this in the article
    The age of the subjects in the training dataset can be seen in lines 83-84. Unfortunately the race of the subjects is not recorded in the patient records available to us. However, we will add information about the time to imaging for these patients! (They are all hyper-acute patients (within 24H))
  2. We do not have a reference standard here, this is part of the novel approach we are suggesting to automate the FLAIR segmentation. A justification is given in lines 132-135. We also perform an ablation study, where we remove this component in the results section (lines 201-207) and show that removing this criteria/component worsens the agreement with the manual segmentations
  3. This is a poor formulation on our part which we will correct in the revised article. The blue dots corresponds to those patients that are assessed to be no-mismatch patients at hospital submission - since the assessment is binary and manually done, it is unlikely that the ratio should be exactly one, just like the patients assessed to be mismatch do not all have 0 ratio. - We will clarify this in the revision

Reviewer 2 Report

Comments and Suggestions for Authors

Thank you for submitting the good report of DWI.FLAIR mismatch detecting by AI and its suggestion that the mismatch is continuous character.

Material and Method

Can we use diagnostic evaluations by physicians such as neurosurgeons and stroke neurologists? Please state this in the Limitation, as it is not necessarily diagnosed by a radiologist.

You used U-net, but its algorithm is a bit old. How about using other algorhithm like transformer?? Discuss this.

Discussionn

Please refer this article about rehabilitation after stroke and mention the prognosis prediction by AI. Is your AI can contribute to the diagnosis of the prognosis of ischemic stroke? PMID: 34466308 

Author Response

Dear reviewer, thanks for the feedback to our article.

We have the following comments:

  • All the evaluations in the paper are made by neuro-radiologists, which are radiologists specialized in neuro imaging. In the Hospital system where the research has originated, segmentations and DWI/FLAIR mismatch assessment would normally be done by neuro-radiologists as these are the most qualified / trained people to do so.
  • The u-net could be swapped for a transformer model (or any other method capable of segmenting DWI stroke lesions) - we will clarify and discuss this in the paper. We have chosen to use the U-net model as it was available to us at the time of writing and have shown reliable results in terms of performance on real world data. 
  • Thank you for the interesting article. We will look into and add to the discussion that this measure might be able to predict longer term recovery in ischemic stroke patients

Reviewer 3 Report

Comments and Suggestions for Authors

This study presents a segmentation method for ischemic stroke patients, which shows promise in determining patient eligibility for recombinant tissue-type plasminogen activator (r-tPA) treatment. Following concerns need to be addressed:

1. The computation time for the developed system has not been reported. The authors should include this information.
2. Further details of the proposed method are required. The authors are advised to incorporate a model diagram and clearly articulate the novel aspects of their method. Additional information should be provided in the methodology section.
3. The Results section lacks depth. The authors are encouraged to conduct some kind of ablation study on the proposed method. Although, it is good that statistical tests were performed.
4. The abbreviation DWI should be expanded, and an appropriate reference should be cited. The authors need to justify their choice of DWI in the context of their research.
5. Please add comments on reproducibility code etc.
6. Authors may discuss how their method compares with existing segmentation works.
7. Some works also explore medical segmentation, such as - Singh et al. (2023). Latent Graph Attention for Enhanced Spatial Context. - Thomalla et al. (2011). DWI-FLAIR mismatch for the identification of patients with acute ischaemic stroke within 4· 5 h of symptom onset (PRE-FLAIR): a multicentre observational study.
8. Please make sure all references follow uniform format.

Author Response

Dear reviewer, 

Thank you for the feedback to our article! 

We have the following comments;

  1. This is a very good point, we will add it to the revised version (The algo runs in almost real time ~5-10 seconds, provided the DWI image is obtained before the FLAIR image.)
  2. We will add a simple explanation of the method/algorithm to the revision, most likely in the form of a flowchart / model diagram
  3. It is unclear what is sought after more here. We perform an ablation study by removing the multiplier component of the algorithm, resulting in worse performance. Furthermore we test the robustness of the method by varying the parameters to show generalization. Can you expand on what you think is missing in this part?
  4. This comment is also unclear to us. We use DWI as it is the primary sequence for identifying infarction / ischemic stroke on MRI. Furthermore, as explained in the introduction, the DWI / FLAIR mismatch criteria suggested by Thomalla et al, has become the gold standard (atleast in western europe / the US) for dealing with unknown onset patient treatment - would adding something along these lines to the article address the DWI justification?
  5. We will add this along with the model diagram / flow chart from point 2
  6. & 7. While it would be ideal to compare our method to pre-existing methods, we have only found the one method in the paper that actually segments FLAIR hyperintensities in strokes. While much work is done, it is focused around automatic DWI/FLAIR mismatch classification or classification of whether the patient is within 4.5 hours of onset, which does not circumvent the binary labelling problem described in the introduction.  Thomalla et al. (2011) for instance predicts whether the patients are within 4.5 & 6 hours, but do not directly segment FLAIR hyperintensities automatically. While many automatic segmentation methods exists, for instance the one you cited (Singh et al. (2023). Latent Graph Attention for Enhanced Spatial Context.) no open-use FLAIR hyperintensity segmentation dataset exists to our knowledge, meaning no ground truth labels are available for comparison or training such models. This is also why we have released the ISLES FLAIR segmentation labels for open use. While we could re-train a method like Singh et al, on our ISLES labels, the work required would be substantial and it is unlikely that the amount of data is enough for the method to perform well, considering we have less than 50 images, compared to the 5000 images used in the paper.
  7. .
  8. We will re-check the references to make sure they are correct / uniform

Again, thanks for the substantial and detailed feedback

Round 2

Reviewer 3 Report

Comments and Suggestions for Authors

I thank the authors for incorporating the suggestions. I understand that it will be extra work to do comparison with Singh et al. (2023). Latent Graph Attention for Enhanced Spatial Context, however, it should be included in the literature review section.

Author Response

Thanks for the reply.

We have included 5 references to the introduction as well as further expanded on why a deep learning model approach is not feasible for the problem described in the paper (line 47-55)